statistical physics/systems theory/computer modelling and simulation

online social network, information diffusion, cascade prediction

**Author for correspondence:**
Xiao-Ke Xu
e-mail: xuxiaoke@foxmail.com

# Why cannot long-term cascade be predicted? Exploring temporal dynamics in information diffusion processes

Ren-Meng Cao[1,2], Xiao Fan Liu[3] and Xiao-Ke Xu[1]

[1]College of Information and Communication Engineering, Dalian-Minzu University, Dalian, People's Republic of China
[2]Institution of Science and S. and T. Management, Dalian University of Technology, Dalian, People's Republic of China
[3]Web Mining Laboratory, Department of Media and Communication, City University of Hong Kong, Hong Kong SAR, People's Republic of China

 R-MC, 0000-0001-6328-5814; XFL, 0000-0001-8342-4623;
X-KX, 0000-0002-9148-3145

Predicting information cascade plays a crucial role in various applications such as advertising campaigns, emergency management and infodemic controlling. However, predicting the scale of an information cascade in the long-term could be difficult. In this study, we take Weibo, a Twitter-like online social platform, as an example, exhaustively extract predictive features from the data, and use a conventional machine learning algorithm to predict the information cascade scales. Specifically, we compare the predictive power (and the loss of it) of different categories of features in short-term and long-term prediction tasks. Among the features that describe the user following network, retweeting network, tweet content and early diffusion dynamics, we find that early diffusion dynamics are the most predictive ones in short-term prediction tasks but lose most of their predictive power in long-term tasks. In-depth analyses reveal two possible causes of such failure: the bursty nature of information diffusion and feature temporal drift over time. Our findings further enhance the comprehension of the information diffusion process and may assist in the control of such a process.

# 1. Introduction

The study of information diffusion in online social networks has practical values in various domains, such as advertisement

placement [1,2], content recommendation [3,4] and infodemic controlling [5,6]. One of the essential research topics in information diffusion is to predict the popularity of a piece of information (i.e. its diffusion scale or cascade size) based on early-stage spreading dynamics [7,8]. Existing prediction methods can be roughly divided into two main paradigms. One is to use the point process models to simulate popularity dynamics [9–11]. The other is to train predictive machine learning (ML) models with historical data [12–14]. With the cost of data acquisition decreasing, researchers are inclined to ML models with features extracted from large-scale datasets to achieve high prediction accuracy and explore the factors that drive information propagation [13,15].

The popularity of a piece of online information is influenced by many factors, such as its content and the social influence of the posting user and the ones who further spread it. To estimate long-term popularity, existing studies built their methods upon the correlation between early and future popularity [16–18]. However, the effect of these factors may change dynamically with the diffusion process, making long-term popularity prediction a challenging task [7]. For example, although early popularity dynamics have been proved to be a strong predictor for future popularity, the bursty nature of online information breaks the correlation between early and future popularity in the long term [19]. A typical burst scenario in online information diffusion is when a tweet has a low activity (e.g. the number of retweets) in the early diffusion process but suddenly gains its popularity owing to the retweet by a key opinion leader. In this case, the ML model features constructed based on the early retweeting dynamics will be invalid in the prediction task.

Recent studies on long-term cascade prediction have made great efforts to tackle the above challenge. Except for historical popularity series, content features such as the number of hashtags or URLs in a tweet [20], the language, text length and the sentiment of a tweet [21], original poster features like the number of followers, the number of past posts, and the average popularity of past posts [20,22], and the user interactions including the user comments, likes, and dislikes [23] are all shown to be correlated to content popularity. Besides, many scholars model the spread of a piece of information as a stochastic event under some specific assumptions about the popularity evolution. For example, Shen *et al.* [24] proposed a generative probabilistic model using a reinforced Poisson process to explicitly model the process through which individual items gain their popularity. Zhao *et al.* [25] modelled information cascade as a Hawkes process with two components, i.e. the human reaction time of sharing and the time-varying post infectiousness of tweets. Other studies assumed that information popularity is a combination of endogenous factors such as user interactions within the system and exogenous factors such as external events [26,27]. On the whole, although these methods achieve acceptable prediction performance, some factors that make long-term cascade prediction still need to be further discussed and exploited.

In this study, we examine the predictive power (and the loss of it) of the most commonly used features in both short-term and long-term information cascading predictions. Specifically, we consider a Twitter-like online social networking service as a typical example of information diffusion and adopt a best-practice ML method to predict the cascade size, i.e. the number of people who retweeted a original post. Four groups of early-stage retweeting patterns that are known to be effective in the prediction tasks are used to train an eXtreme Gradient Boosting (XGBoost) model, including following network structure [28,29], retweeting network structure [12,30], early retweet popularity [21,31] and tweet content features [15,20].

We compare the predictive power of each group of features in both short-term and long-term prediction tasks as well as the loss of indicativeness of these features. For the group of features that account for the largest loss of predictive power, we further scrutinize the reason that probably caused such failure, by summarizing typical cases from the data. Our findings further enhance the comprehension in the information diffusion process and may assist in the control of such process. Besides, our research method can not only be applied to the information dissemination of social networks such as Twitter, Facebook and Weibo, but also other fields such as book sales recommendation [32] and citation impact of scientific papers [33].

The rest of the paper is organized as follows: §2 describes the data we used; §3 describes the features we used and analyses the prediction results of various features; §4 explores some temporal dynamics that make cascade size difficult to predict; and finally, §5 is the conclusion.

## 2. Data description and preprocessing

In this study, we use a large-scale dataset of Sina Weibo (i.e. a similar online social network platform to Twitter) provided by DataCastle.[1] Users on Weibo can post tweets and share other users' tweets by

'retweeting'. A user's original tweets and retweets will appear on its followers' timelines. The dataset contains three parts:

— *original tweets:* the tweet identity (ID), posting user's ID, release time and content for 30 010 original tweets. These tweets were posted between 1 January 2015 and 20 May 2016;
— *retweets:* the original tweet's all retweets, including retweeter ID and retweeting time. There are 15 975 387 retweets from 7 445 377 users; and
— *following relationship:* the lists of users that each of the posting users and retweeters followed, including 7 977 942 users and 700 434 403 following relations.

# 3. Predicting information diffusion scale

In this section, we adopt a best-practice method to predict the information diffusion scale. First, we extract features of the users' following and retweeting network, tweet content and early popularity dynamics. Then, we use a robust ML model (i.e. XGBoost) to forecast the cascade sizes of the original tweets based on these features. Finally, the prediction performance and loss of it for each group of features are analysed and compared.

## 3.1. Features for machine learning model training

We have developed an extensive list of features, including both previously reported and newly coined ones, for ML model training (table 1). These features can be grouped into four categories: the following network structure, retweeting network structure, early-stage popularity dynamics and tweet content.

### 3.1.1. Following network structure

On online social media platforms, people can *follow* the ones that they are interested in. Based on the following relationships among users, a network $G_F = (U_F, E_F)$ can be constructed, where each user $u \in U_F$ is represented as a node and each directed edge $e(u_s, u_t) \in E_F$ from $u_s$ to $u_t$ is established when $u_t$ follows $u_s$. This network not only provides pathways for information diffusion but also affects the size of the information cascade. For example, prior studies [20,35] have shown that users with a large number of followers could lead to large retweet cascades.

Five network structural features are constructed to measure the user influence in the following network: *out_degree_F, in_degree_F, all_degree_F, bi_degree_F* and *pagerank_F*, where *out_degree_F* and *in_degree_F* were both studied in previous works [15,20,28,30].

### 3.1.2. Retweeting network structure

Retweeting is a key mechanism of information diffusion and a specific manifestation of information popularity [36]. Maleewong [37] found that the retweets from active users (who have stable and frequent participation in posting and retweeting) could receive a large number of retweets. For each original tweet, a retweeting network $G_R = (U_R, E_R)$ based on the retweet data in the first 60 min is constructed. In $G_R$, each user $u \in U_R$ is represented as a node and each directed edge $e(u_s, u_t) \in E_R$ from $u_s$ to $u_t$ is established when $u_t$ retweets one of the original tweets or retweets from $u_s$.

Five network structural features are constructed to measure the user influence in the retweeting network: *out_degree_R, in_degree_R, all_degree_R, bi_degree_R* and *pagerank_R*, in which *pagerank_R* was studied in previous works [13,30].

### 3.1.3. Early retweet dynamics

The early-stage retweet time series contains rich temporal information. Szabo & Huberman [16] found a strong linear correlation between the log-transformed popularity at early and later periods in Digg and Youtube. Hu *et al.* [14] quantified the temporal features from three dimensions: the overall level of the popularity, increasing or decreasing trends, and cyclicity. To capture the trends and fluctuations of cascade sizes over time, four types of temporal features are constructed. The detailed descriptions of the features are as follows:

[1]www.pkbigdata.com.

**Table 1.** Four categories of features for cascade prediction.

| features | description | representation |
|---|---|---|
| following network structure (FNet) | number of people who follow the posting user | *out_degree_F* [15,20,28,30] |
| | number of people the posting user follows | *in_degree_F* [15,20,28,30] |
| | number of people who follow or are followed by the posting user | *all_degree_F* |
| | number of people who follow and are followed by the posting user | *bi_degree_F* |
| | PageRank [34] centrality in $G_F$ | *pagerank_F* |
| retweeting network structure (RNet) | number of people who retweet the posting user's tweets | *out_degree_R* |
| | number of people retweeted by the posting user | *in_degree_R* |
| | number of people who retweet or are retweeted by the posting user | *all_degree_R* |
| | number of people who retweet and are retweeted by the posting user | *bi_degree_R* |
| | PageRank centrality in $G_R$ | *pagerank_R* [13,30] |
| early popularity (early) | cascade sizes in the observation period organized in 1 min intervals | *cascade* |
| | burstiness of popularity in the early period | *burstiness* |
| | stability of popularity in the early period | *stability* |
| | the release time of the original tweet | *release_time* |
| tweet content (content) | whether the original tweet has URLs | *URL* [15,20] |
| | whether the original tweet has hashtags | *hashag* [15,20] |
| | whether the original tweet has photos | *photo* [15] |
| | whether the original tweet contains '@user name' terms | *mention* [15,20] |
| | topics category of the original tweet | *contentgory* [13] |
| | text length of the original tweet | *wordlength* [15,28,30] |

— cascade: the cumulative time series of cascade sizes organized in 1 min intervals in the observation period. Note that cascade = {cascade$_1$, cascade$_2$, ... , cascade$_t$}, where $t$ represents the length of the observation period in minutes, and cascade$_t$ denotes the number of people who retweeted the original post when it was spread at time $t$;

— burstiness: this index measures the burst of the popularity time series in the observation period, which is defined as

$$burstiness = \frac{peak}{cascade_t},$$ 
(3.1)

where peak represents the largest net increase of the retweet time series in the observation period;

— stability: this index measures the stability of the popularity time series in the observation period, which is defined as the ratio of the standard deviation $\delta$ of the time series *cascade* to the mean $u$, that is,

$$stability = \frac{\delta}{u},$$ 
(3.2)

— *release_time*: the release time of the original tweet by the hour. As an example of a tweet posted at 18.30, the release time of the tweet is represented as 18.5 h.

### 3.1.4. Tweet content

We use the term frequency-inverse document frequency algorithm [38] to extract the feature vectors of content text and used the latent Dirichlet allocation (LDA) topic model [39] to obtain the topic distributions of each original tweet. Apart from text, multimedia and hyperlinks can be used to convey information, which increases the diversity of information expression and affects the popularity of information. For example, Suh *et al.* [20] found that the numbers of hyperlinks and hashtags have a strong correlation with retweet likelihood by using principal component analysis (PCA). Zhao *et al.* [40] found that tweets with multimedia information have a greater number of retweets and a longer life span than plain text messages. Besides, the topics, length and release time of original tweets also affect retweet probability [13,28,41].

Considering the influence of contents of original tweets on the retweet likelihood, six features are extracted from the tweet content, where the feature of *contentgory* was mentioned in [13]. The rest were mentioned in [15,20]. The detailed descriptions of the features are as follows:

— *contentgory*: the topics category of an original tweet extracted by the LDA topic model;
— *wordlength*: the text length of the original tweet;
— *URL*: whether the original tweet has URLs;
— *hashtag*: whether the original tweet has hashtags;
— *photo*: whether the original tweet has photos; and
— *mention*: whether the original tweet contains *@user name* terms.

## 3.2. Prediction method, evaluation metric and results

The prediction problem is defined as follows: given the observation of retweeting dynamics of an original tweet $m$ after being posted for a period $t$, predict the cascade size of the original tweet at time $t + \Delta t$. The cascade size at a certain time can be viewed as a continuous value; thus, we treat the prediction task as the classical regression problem. The XGBoost model [42] is used as the training algorithm.

We use the root mean square error (RMSE) [43] to evaluate the predictive performance of our model. The RMSE measures the deviation between real and predicted values. The smaller the value of RMSE, the better predictive performance we achieve. Let $N_m(t)$ be the real cascade size of the original tweet $m$ at time $t$ and $\hat{N}_m(t)$ be the predicted cascade size of $m$ at time $t$. For a dataset with $M$ original tweets, the prediction error in RMSE at time $t$ is defined as

$$\text{RMSE}(t) = \sqrt{\frac{1}{M}\sum_{m=1}^{M}(\hat{N}_m(t) - N_m(t))^2}. \tag{3.3}$$

We consider two prediction scenarios: short-term and long-term prediction. In the short-term prediction, we take the first 60 min of retweeting dynamics as training samples to predict the information cascade size at the 75th min. In the long-term prediction, we take the same training samples to predict the information cascade at the 4380th min. For each target, we retrain our prediction model; 26 955 Weibo original tweets posted in 2015 are randomly divided into two groups: 80% for training and 20% for testing.

Figure 1 shows the prediction performance and the loss of the predictive power of different feature groups. In the short-term prediction, features of early popularity dynamics outperform other groups of features. In the long-term prediction, the RMSEs of all feature groups substantially increase, and the most significant growth is observed in early popularity dynamics, despite that it is still the most informative group of features. In summary, features of early popularity dynamics not only are the key predictors of cascade size but also lose their predictive power most severely from short-term to long-term prediction.

# 4. Why are long-term predictions difficult?

The results from the last section showed two apparent trends. First, the accuracy of the prediction tasks drops sharply when the prediction target becomes remote. Second, the most informative features for short-term prediction (i.e. the early popularity dynamics) lose their predictive power most significantly in the long-term prediction tasks. Considering that the structural properties of the

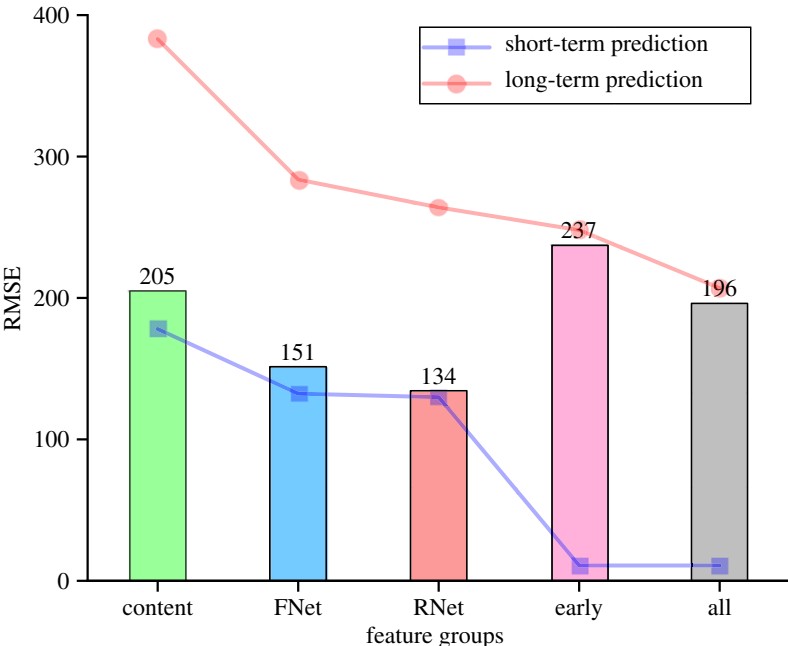

**Figure 1.** Prediction performance and the loss of predictive power of different feature groups in short-term and long-term cascade prediction. The blue and red lines show the prediction performance of each group of features and all features in the short-term and long-term prediction tasks, respectively. The bar chart shows the growths of RMSEs from short-term to long-term prediction targets. RNet denotes the retweeting network features, FNet denotes the following network features, early denotes the early popularity dynamics, Content denotes the features of original tweets' content features, and all denotes the collection of all the 79 features.

following and retweeting networks and the tweet content features have very little predictive power, we conjecture that the difficulty of predicting long-term information cascade lies in the decay of the auto-correlation between the early and future popularity. The possible factors that break such a link may come from many different aspects. Several previous works showed that the bursty nature of online content popularity evolution and the dynamic change of user online activity can break the correlation between early and future popularity [11,12,19]. On the basis of prior works, we further explore the unpredictable events in information diffusion processes and summarize several common scenarios in which the actual tweet popularity largely diverges from the predicted scales.

## 4.1. The bursty nature of popularity evolution in the information diffusion process

Burst is a common phenomenon that exists in many real systems. It can be characterized by short periods of intense activity followed by long times of no or reduced activity [44,45]. The appearance of bursts can break the correlation between early-stage popularity and long-term popularity in information diffusion. For example, two original tweets with similar early-stage popularity can either experience a burst in the future propagation or not (figure 2a).

Burst can be characterized by peaks in the diffusion curves [46,47]. Here, we design a peak detection algorithm. Given an equally spaced time series that describes the net growths of cascade size of an original tweet $C(t) = \{c(t_i), c(t_{i+\Delta x}), \ldots, c(t_{i+n\Delta x})\}$, a peak is defined as the local maximum of the time-series segment within a fixed window size $w$, namely peak $= \max\{c(t_{j-w}), c(t_{j-w+1}), \ldots, c(t_{j+w})\}$. Besides, the height of a peak is at least $k$ times larger than the mean time series $\bar{C} = (1/n)\sum_{l=0}^{n} c(t_{i+l\Delta t})$. The difference in height between any two adjacent peaks should be greater than $m$.

In practice, we set $k = 4$, $w = 2$ and $m = 10$ and perform a peak detection for all original tweets in the testing set. The original tweets with one peak in their diffusion curves account for 33.4%, those with more than one peak 52.6%, and no peaks 14.0%. The result demonstrates that most original tweets in the testing set exhibit one or more bursts in the information diffusion process.

To test the impact of burstiness to the cascade size predictability, we compared the prediction errors between original tweets that experienced strong, weak and no burst in the cascade. An original tweet is considered to have experienced a strong burst if its largest peak height is higher than the average height of all peaks of all the original tweets' cascades. Other original tweets with at least one burst are

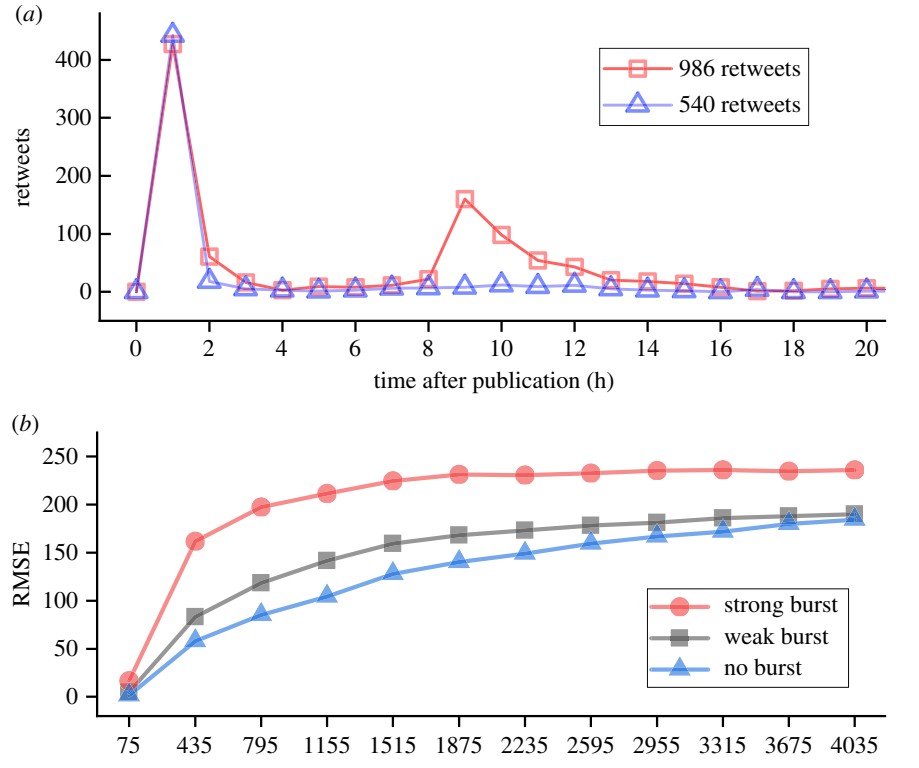

**Figure 2.** (*a*) Burst in the diffusion process. Two original tweets with similar early-stage popularity can either have a burst in the future propagation (red curve) or not (blue curve). (*b*) The prediction performance of XGBoost models for original tweets that experienced strong, weak, and no burst in their cascades. Higher RMSE reflects lower prediction precision. Cascades with strong bursts are the most difficult to predict while those with weak bursts and no burst are gradually and comparatively easier to predict.

considered to have experienced weak bursts. In the end, 38% of the original tweets are tagged with strong bursts, 48% with weak bursts and 14% with no burst.

The prediction performance, i.e. RMSE, for original tweets that experienced strong, weak and no burst at different times in their cascades are shown in figure 2*b*. One model is trained and tested for each group of samples, by randomly selecting 80% of data for training and the remaining 20% for testing. Evidently, the original tweets that experienced strong bursts have the lowest prediction performance, while the less severe or fewer bursts an original tweet experiences, the easier it is to predict its cascade size. Therefore, the strength of the bursts indeed leads to the (un)predictability of information cascades.

## 4.2. Hard-to-predict dynamics in multi-peak propagations

The number of peaks in the information cascades may also exert influences on the prediction performance. Particularly, multiple bursts in the information diffusion increase the uncertainty of cascade propagation, making long-term popularity prediction more difficult. Figure 3*a* shows the real and the predicted diffusion curves of a typical original tweet with multiple peaks in the diffusion curve. The real cascade size increases to about 150 at the end of the first hour and then experiences two large and rapid increases, with the cascade size rising from less than 200 to about 1100. However, the predicted cascade size grows in a relatively flat trend, from less than 200 to about 700 steadily, resulting in a large error between the empirical value and predictions.

To further explain the cause of the large prediction error of the cascade size of the original tweet in the above experiment, the cascade propagation tree of the original tweet and its net growth curve are drawn in figure 3*b,c*. In the retweeting network, the red, yellow and blue nodes represent the users who retweet the post, corresponding to the three peaks in the cascade size curve. In the first peak, node A contributes to the major number of retweets. The second peak is caused by two influential users (i.e. nodes B and C). The two influential users spread the information cascades to their followers respectively, forming two new clusters. The third peak occurs because the two influential users (i.e. nodes B and C) promote the spread of information cascades once again. Also, two new influential users (i.e. nodes D and E) appear and further improve the diffusion of the information cascades.

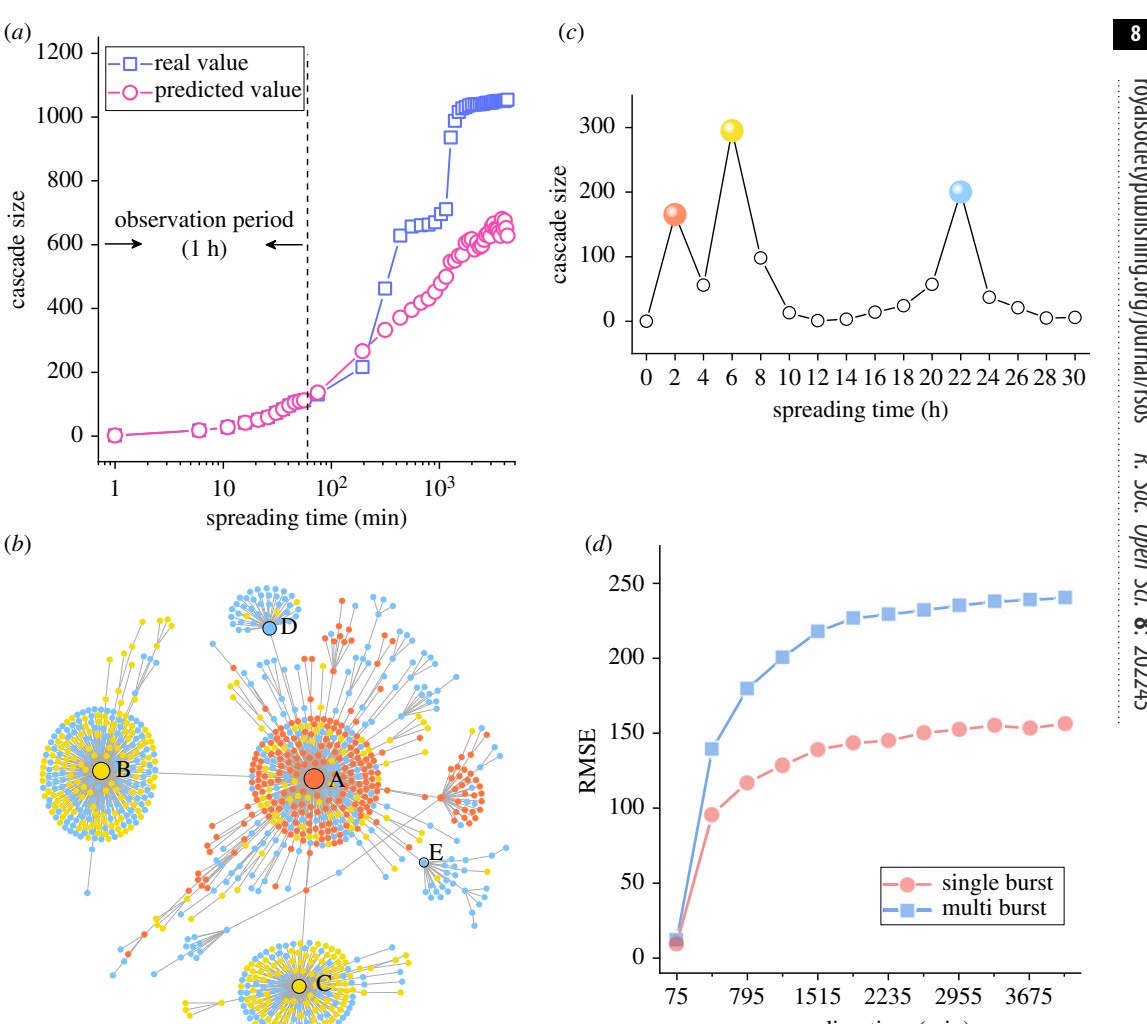

**Figure 3.** (a) Cumulative growth curves of the real and predicted values of an original tweet with multi-peak propagation. The red curve is the predicted values starting from the 75th min with 120 min increments. For each prediction target, we re-train the model. (b) The retweeting network of a multiple-burst tweet. (c) The cascade size time series of multiple-burst tweet information cascades. The nodes with labels are the most influential users in the retweeting network. (d) The prediction performance of XGBoost models for original tweets that experienced multiple peaks or a single peak in their cascades, respectively. Higher RMSE reflects lower prediction precision. Prediction performance for original tweets with multiple peaks in their cascades is lower than those with a single peak.

The unexpected retweet by a key opinion leader in the online social network may cause unforeseeable new cascades of the original tweets. These random events happen at a relatively low probability, and therefore diverge from the most probable scenarios modelled by the ML algorithms and cause large prediction errors.

Similar to the previous section, we stratified the original tweets that experienced at least one burst into those which experienced only one burst and those which experienced multiple bursts. One model is trained and tested for each group of samples, by randomly selecting 80% of data for training and the remaining 20% for testing. Based on the previous result of peak detection, the original tweets with one peak in their cascade account for 52% while the tweets with multiple peaks occupy 48%. Figure 3d shows that it is easier to predict the cascade size for original tweets that experienced only one peak in their cascades. This finding confirms that the complex burstiness in information cascades hinders the predictability of the cascade process.

## 4.3. The variability of the original time of posting

We characterize the hourly variations of user activity by the number of retweets at different moments in a day. As shown in figure 4a, user activity changes over time; that is, users are more inclined to post

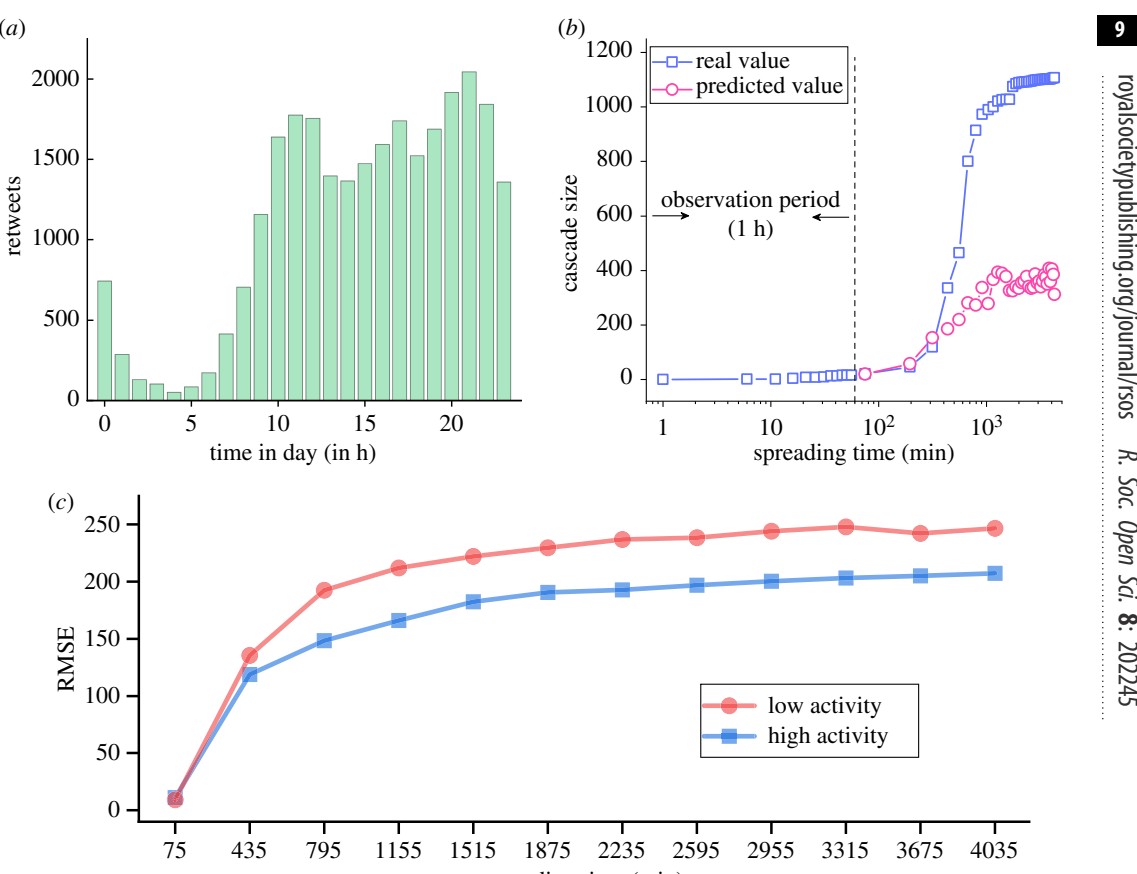

**Figure 4.** (*a*) The number of retweets at different moments in the day in our dataset. Each bar represents the number of retweets in an hour. (*b*) The real and predicted cascade sizes of an original tweet published around 5.00. The red curve is the predicted values starting from the 75th min with 120 min increments. For each prediction target, we re-train the model. (*c*) The prediction performance of XGBoost models for original tweets that are posted at different time in a day. Higher RMSE reflects lower prediction precision. Prediction performance for original tweets posted during high activity period is better than those posted during low activity period.

original tweets and retweet in the daytime rather than at midnight. In this study, we consider 0.00 to 8.00. as the low activity period and 8.00 to 23.00 as the high activity period.

Figure 4*b* shows the real and predicted diffusion curves of an original tweet, which is posted around 5.00. The original tweet receives a small number of retweets after being posted for 1 h. With the passage of time, the tweet information cascades transits gradually to the periods of high user activity. From the 75th to 435th min, the real cascade size of the original tweet increases at a relatively steady rate, and the errors between the real and the diffusion curve are very small. From the 435th to 675th min, user activity reaches a relatively high level, while the predicted cascade size still increases at a relatively slow rate, resulting in a significant prediction error. These results show that the dynamic change of user activity leads to a significant change in cascade growth trends of early-stage and long-term popularity dynamics, which significantly reduces the prediction ability of the features built on the early popularity dynamics.

To further test the difference in predictability for cascades that started in high and low activity periods, respectively, we stratified the original tweets by their posting time into two groups. One model is trained and tested for each group of samples, by randomly selecting 80% of data for training and the remaining 20% for testing. Figure 4*c* shows that it is easier to predict the cascade size for original tweets posted during a high activity period than those posted during a low activity period. This finding confirms that original tweets that are posted during a low activity period may experience unexpected cascading behaviour and diverge from prediction severely.

## 4.4. Feature temporal drifting

Feature drift occurs when the validity of features in the data stream changes over time, requiring ML models to be retrained to ignore irrelevant features and consider the newly relevant ones [48].

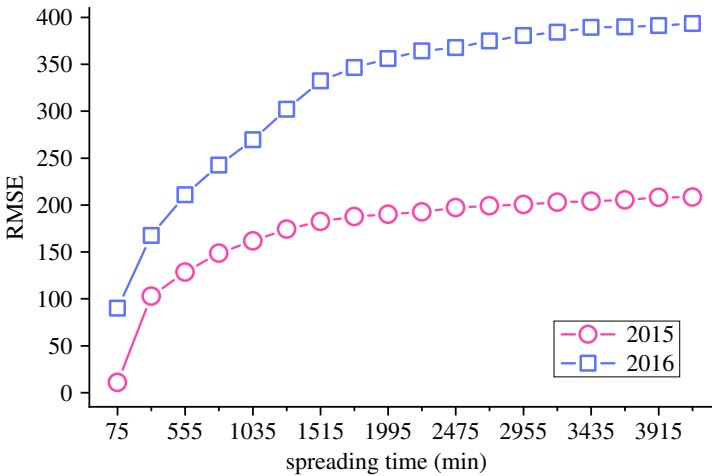

**Figure 5.** The RMSE curves of prediction results based on the testing set from 2015 and 2016.

Barddal *et al.* [49] analysed the impact of feature drift on some ML algorithms and found that the existing ML algorithms did not solve the problem of feature drift. Therefore, when performing an ML task on the data with the feature drift phenomenon, the influence of temporal factors on the prediction results should be considered.

To confirm the existence of feature drift in cascade propagation prediction, a predictive model is constructed and trained based on 10 000 original tweets published in 2015, and 3000 original tweets published in 2015 and 2016 are randomly selected for testing, respectively. There is no overlap between the training data and the testing data. Figure 5 shows the curve of RMSE, the prediction accuracy index, with the growth of spreading time when the prediction is made based on the testing sets of different years. As can be seen from the figure, the accuracy of the prediction result based on the testing data of 2015 is better than that based on the testing data of 2016, and this experimental result proves that the phenomenon of feature drift does exist in the prediction of cascade propagation. Therefore, in the cascaded propagation prediction using ML methods, it is necessary to construct the features with a strong time correlation with the prediction target variables; otherwise, the prediction accuracy will be greatly affected.

## 5. Conclusion

This paper attempts to explore the possible causes of difficulties in predicting information diffusion in the long run. First, we use a robust algorithm (i.e. XGBoost) to build ML models using various features from the early-stage dissemination of original tweets in a popular online social network. Then, we compare the predictive power of different groups of features in the short- and long-term cascade sizes and found that the prediction error increases sharply with the prediction gap. The most predictive features (i.e. early popularity dynamics) in the short-term prediction tasks lose most of their predictive power in the long-term tasks.

In the light that the early-stage information propagation dynamic features lose their predictive power in the long-term prediction tasks, we conjecture that the difficulty of predicting long-term popularity lies in the decay of correlation between early and future popularity. We design a peak detection algorithm to examine the unpredictable events in the information diffusion process and summarize two common scenarios in which the actual information cascades largely diverge from the predicted scales: (i) the burstiness of popularity; and (ii) the dynamic change of user activity. Also, the phenomenon of feature temporal drift reduces the validity of feature engineering we build, making long-term popularity more difficult to predict.

Our findings have two-sided implications for the prediction and control of information diffusion in online social networks. On the one hand, we suggest that there is potentially an upper bound of accuracy when trying to predict future information cascade size using early-stage dynamics only. That is, early popularity does not mean long-term popularity and vice versa. The complex online environment may generate many unexpected factors that can severely affect the information propagation process and the diffusion scale. On the other hand, we suggest that the information diffusion process is generally

controllable. The controllability embodies in two ways: micromanagement during the diffusion process, e.g. the introduction, or the suppression, of a key opinion leader, which can boost an already started diffusion process or prevent undesired future diffusion; and changing the underlying online environment can also alter the information diffusion mechanism, hence information popularity in the new environment.

Data accessibility. Data and relevant code for this research work are available at https://doi.org/10.5061/dryad.44j0zpccv [50].

Authors' contributions. R.M.C., X.F.L. and X.K.X. designed the study. R.M.C. and X.K.X. designed the experiments. R.M.C. analysed the experimental data. X.F.L. and X.K.X. interpreted the results and wrote the manuscript. All authors gave final approval for publication.

Competing interests. We declare we have no competing interests.

Funding. This work was supported by the National Natural Science Foundation of China (61773091 and 61603073), the LiaoNing Revitalization Talents Program (XLYC1807106), and the Major Project of the National Social Science Fund of China (grant/award no. 19ZDA324).

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
