## [Peer Review File · Royal Society Open Science]

Review History

RSOS-202245.R0 (Original submission)

Review form: Reviewer 1

Is the manuscript scientifically sound in its present form?

Yes

Are the interpretations and conclusions justified by the results?

Yes

Is the language acceptable?

Yes

Do you have any ethical concerns with this paper?

No

Have you any concerns about statistical analyses in this paper?

No

Recommendation?

Accept with minor revision (please list in comments)

Comments to the Author(s)

In this manuscript, the author aimed to explore the possible causes of difficulties in predicting information diffusion in the long run. Comparing the predictive ability (and loss of it) of various feature sets, the authors found that early diffusion dynamics are the most predictive ones in short-term prediction tasks but lose most of their predictive power in long-term tasks. Afterward, the authors further dug out two causes that can break the correlation between early and future popularity. Overall, the paper is well-structured and innovative. I recommend publication after some minor revision.

1. In Section 1. Introduction, the author needs to provide more literature about the impacts of the busy diffusion of online content on popularity prediction and a detailed description about how prior studies solve the difficulty in long-term prediction, so as to put forward the research question.
2. I suggest that " This dataset contains an original tweeter ID and a list of user IDs the original tweeter follows" in Section 2 Data description and preprocessing might be revised into "original tweeter followed".
3. In Section 3 Following network structure, "mutual friends" may be revised into "mutual following".
4. In Section 3 Retweeting network structure, I am confused about the description of all_degree_R. In the retweeting network, the relationships between users should be retweet or tweet, not a friend.
5. In Section 3 Early retweet dynamics, the author constructed four types of features instead of three.
6. In Section 3 Early retweet dynamics, "cascade 60" might be revised into "cascade t".
7. In Section 3 tweet content, I am confused about the text "The texts bare the richest information about individual tweets."
8. In Figure 1, there are some typos in the caption, "the blue and short lines" might be revised into "the blue and red lines".
9. In Section 4(a), I am confused about the expression "Bursts can appear with very rare probability and hence break the correlation between early-stage popularity and long-term popularity in information diffusion." In this section, the authors should focus on the effect of burst on the spread of information, not the probability of burst.
10. In Section 4(b), the authors might give a definition of high/low user activity periods.
11. In Section 4(c), the authors might label the key nodes in the network in Figure 6, so as to give a clear description of the information diffusion process. Besides, the description of Figure 6 is not consistent with the information the figure conveys. For example, the text "The third peak occurs because these two influential users further ... followers."

Review form: Reviewer 2

Is the manuscript scientifically sound in its present form?

Yes

Are the interpretations and conclusions justified by the results?

Yes

Is the language acceptable?

Yes

Do you have any ethical concerns with this paper?

No

Have you any concerns about statistical analyses in this paper?

No

Recommendation?

Accept with minor revision (please list in comments)

Comments to the Author(s)

In "Why cannot long-term cascade be predicted? Exploring temporal dynamics in information diffusion processes", Xiao-ke Xu, Ren-meng Cao and Xiao-fan Liu explore long-term and short-term prediction of the popularity of tweets on the social media platform Weibo. They use a machine learning model which bases its predictions on the characteristics of the Weibo tweet, features of the tweet's early retweeting record, and the characteristics of the user who posted the tweet. I have little experience in machine learning or tweet dynamics, and cannot comment on whether the machine learning model and training process are identified unambiguously. To me, the paper seemed well-motivated and well-organised, with conclusions largely justified by the data and supporting analysis. I recommend some small changes before publication.

Conclusions and analysis:

In their current form, some conclusions are not entirely supported by the data and analysis. In such cases, I suggest either adjusting the conclusion or extending the analysis.

(1) It is stated that early popularity dynamics are the fastest to lose their predictive power (p. 7, lines 57-59) and the first to lose their predictive power (p. 8, lines 9-11). In my opinion, statements about the rapidity with which predictive power is lost might be misleading, and should probably be modified. I suggest this because the predictive power is only shown at two extreme prediction horizons, and it is hard to infer how prediction performance varies between these two extremes.

(2) In Section 4 (d) it is shown that a model trained on 10000 tweets from 2015 can predict 3000 tweets from 2015 more accurately than it can predict 3000 tweets from 2016, and this result is used to argue the presence of feature drift. I suggest indicating whether the 2015 training and test data are kept separate. If the 2015 training and test data share tweets then the higher accuracy for 2015 could be partly due to a tendency of in-sample prediction error to be lower. In this case, more discussion or analysis may be needed to support the conclusion about feature drift.

(3) In the final paragraph of the conclusion it is suggested that the information diffusion process is generally controllable. It is suggested that information diffusion can be enhanced by having a key opinion leader retweet the information, which is reasonable. However, the statement about controllability seems to suggest the existence of mechanisms to either enhance or suppress information diffusion. Therefore, I suggest also indicating a way in which information diffusion could be suppressed or adjusting the statement about controllability.

Stylistic and typographical points:

The paper contains a small number of typographical errors, and at points it might be possible to enhance its style or clarity.

(5) I found the following sentences (spanning two paragraphs) in the introduction jarring: "In this case, the features constructed based on the early retweeting dynamics will be invalid in the prediction task.

Why does such failure happen?"

I found these sentences confusing because they appear to involve answering the main question of the paper (Why does long term tweet popularity prediction fail?) immediately *before* the question is asked. I suggest modifying or reordering at least one of these two sentences or paragraphs.

(6) I suggest defining "cascade size" and "retweet tree" in the context of tweets before or as the terms are first used (currently, p. 4, lines 15-17). I suggest including a reference for PageRank where the centrality measure is first mentioned (currently, p. 5, line 51). I suggest defining "publisher" before or as the term is first used (currently, p. 5, line 51).

(7) Page 5, line 51: For consistency with the preceding four lines, I suggest changing "publisher" to "posting user" (assuming that the meaning of the terms is the same).

(8) Page 6, line 12: For clarity, I suggest changing "where t represents the length of the observation period." to "where t represents the length of the observation period in minutes."

(9) Page 6, line 57: Please clarify the connection between cascade size cascade_t at time t and number $N_m(t)$ of retweets of tweet m at time t.

(10) Figures 4-5: I suggest changing "Observarion" to "Observation". I also suggest checking that the caption and legend are consistent – the legend and caption respectively indicate that the blue and red curve show the real values.

(11) Figure 6: I suggest concluding the caption with a full stop.

(12) Page 12, lines 19-20: Please check this definition of feature drift: "Feature drift occurs when the validity of features in the data stream changes over time, leading the ML algorithms to ignore irrelevant features and consider the newly relevant ones [39]." This seems to suggest that feature drift refers to a self-updating skill possessed by machine learning models. I would have found the following definition of feature drift more intuitive: "Feature drift occurs when the validity of features in the data stream changes over time, requiring ML models to be retrained to ignore irrelevant features and consider the newly relevant ones [39]."

Final comment:

(13) The authors hypothesise that burstiness and the variability of the original time of posting are two sources of long term unpredictability. This is reasonable, and it is shown that cascade time series are bursty and that the tweet rate varies with time of day. The hypotheses are not proved, but are also not presented as having been proved, so this does not invalidate the conclusions of the paper. However, in my opinion, the paper would be enhanced by extending the analysis slightly to test these hypotheses about the sources of long term unpredictability. The hypotheses could be validated, for example, by examining how prediction performance varies with burstiness or the original time of posting. Perhaps more robustly, the authors could instead train and test one model on time series with burstiness above a certain threshold, train and test another on time series with burstiness below the threshold, and compare the performance of the two models. Analogously, the authors could train and test models using only data from tweets first posted at different intervals of the day (for example, midnight – 6am, 6am – 12nn, 12nn – 6pm, 6pm-midnight), and compare the performance of the models restricted to different intervals.

Review form: Reviewer 3

Is the manuscript scientifically sound in its present form?

Yes

Are the interpretations and conclusions justified by the results?

Yes

Is the language acceptable?

Yes

Do you have any ethical concerns with this paper?

No

Have you any concerns about statistical analyses in this paper?

No

Recommendation?

Reject

Comments to the Author(s)

Report on "Why can't long-term cascade be predicted? Exploring temporal dynamics in information diffusion processes" by R-M Cao, X-F Liu, X Xu

The manuscript demonstrates problems encountered with long-term prediction of Twitter like information networks. The authors demonstrate that short-term prediction of information diffusion in such networks is a manageable task but outside a pre-defined prediction window the machine learning algorithm fails to give reasonable estimates for the retweet popularity.

The manuscript is well written, contains the necessary information and contains informative as well as good designed figures. Different retweet scenarios are well illustrated and while I am not the biggest fan of the bullet like listing (see two lists on page 3 and 5), I believe this is within the freedom of the authors.

The methodology developed is based on the popular XGBoost algorithm and RMSE is used as a goodness measures of the ML algorithm. As a training set 80% of the data set (Weibo data from 2015) is used and the remaining 20% are the test data. In principle the result shown is not surprising. XGBoost is a tree based model and such methods partition the input space of any given problem in a way that makes them largely unable to extrapolate target values beyond the limits of the training data when used for predictions. Given that the burst dynamics reported on around figure 2 is a rare event, it is not surprising that a tree-based model fails to predict it. Given that this is the main point of the manuscript and it is a well known fact in data science, I do not rate the novelty of this manuscript highly.

The manuscript in addition focusses on a body of research dealing with the analysis of twitter-like data. I do believe that this is a too narrow view and research on prediction in social systems like for example Sornette, D.; Deschates, F.; Gilbert, T.; Ageon, Y (2004). "Endogenous Versus Exogenous Shocks in Complex Networks: an Empirical Test Using Book Sale Ranking". *Physical Review Letters*. 93 (22): 228701. doi:10.1103/physrevlett.93.228701 would give a different interesting perspective to the research area.

Decision letter (RSOS-202245.R0)

Dear Dr Xu

The Editors assigned to your paper RSOS-202245 "Why can't long-term cascade be predicted? Exploring temporal dynamics in information diffusion processes" have now received comments from reviewers and would like you to revise the paper in accordance with the reviewer comments and any comments from the Editors. Please note this decision does not guarantee eventual acceptance.

Please submit your revised manuscript and required files (see below) no later than 21 days from today's (ie 15-Jun-2021) date. Note: the ScholarOne system will 'lock' if submission of the revision is attempted 21 or more days after the deadline. If you do not think you will be able to meet this deadline please contact the editorial office immediately.

on behalf of Marta Kwiatkowska (Subject Editor)
openscience@royalsociety.org

Associate Editor Comments to Author:
Comments to the Author:

Thank you for your patience. Given the commentary from the reviewers, we'd like you to revise the paper. While two of the reviewers are broadly positive towards your work, they nevertheless recommend a range of tweaks that will improve your work and need to be taken into account before the paper may be accepted for publication. The third reviewer notes that there are many

positive aspects to your work, but highlighted that they are concerned at the lack of novelty in the manuscript. While RSOS tries not to make assessments based solely on novelty and impact, we do ask authors to make clear how their work contributes to the existing body of literature and, therefore, how it is meaningful in the wider context the research area.

Given the concerns raised by the third reviewer would not necessarily constitute sufficient grounds to reject (especially, when the other reviewers are comparatively positive), we would like you to do the following:

- 1) Carefully read through and consider the reports you have received from the referees;
- 2) Carefully and thoroughly revise your paper to respond to their commentary - you should pay particular attention to the comments of the third reviewer to ensure that you more carefully situate your work in the existing literature and how your work advances the field: it will not be enough to simply reiterate your findings but you will need to explain how these fit into and develop the field.
- 3) Provide a point-by-point response to each of the comments of the reviewers - clearly outlining how you have responded in the manuscript. If there are comments you have opted not to respond to, you should also make this clear, but bear in mind that the editors may invite one or more of the original reviewers to reassess the paper. If the reviewers and editors are not satisfied that your work is ready for publication the paper may be rejected (if there is any ambiguity here, further reviewers may be invited).

We wish you every success in completing these tasks, and we'll look forward to receiving the revision in due course.

Reviewer comments to Author:

Reviewer: 1

Comments to the Author(s)

In this manuscript, the author aimed to explore the possible causes of difficulties in predicting information diffusion in the long run. Comparing the predictive ability (and loss of it) of various feature sets, the authors found that early diffusion dynamics are the most predictive ones in short-term prediction tasks but lose most of their predictive power in long-term tasks. Afterward, the authors further dug out two causes that can break the correlation between early and future popularity. Overall, the paper is well-structured and innovative. I recommend publication after some minor revision.

1. In Section 1. Introduction, the author needs to provide more literature about the impacts of the busy diffusion of online content on popularity prediction and a detailed description about how prior studies solve the difficulty in long-term prediction, so as to put forward the research question.
2. I suggest that " This dataset contains an original tweeter ID and a list of user IDs the original tweeter follows" in Section 2 Data description and preprocessing might be revised into "original tweeter followed".
3. In Section 3 Following network structure, "mutual friends" may be revised into "mutual following".
4. In Section 3 Retweeting network structure, I am confused about the description of all_degree_R. In the retweeting network, the relationships between users should be retweet or tweet, not a friend.
5. In Section 3 Early retweet dynamics, the author constructed four types of features instead of three.
6. In Section 3 Early retweet dynamics, "cascade 60" might be revised into "cascade t".
7. In Section 3 tweet content, I am confused about the text "The texts bare the richest information about individual tweets."

8. In Figure 1, there are some typos in the caption, “the blue and short lines” might be revised into “the blue and red lines”.
9. In Section 4(a), I am confused about the expression “Bursts can appear with very rare probability and hence break the correlation between early-stage popularity and long-term popularity in information diffusion.” In this section, the authors should focus on the effect of burst on the spread of information, not the probability of burst.
10. In Section 4(b), the authors might give a definition of high/low user activity periods.
11. In Section 4(c), the authors might label the key nodes in the network in Figure 6, so as to give a clear description of the information diffusion process. Besides, the description of Figure 6 is not consistent with the information the figure conveys. For example, the text “The third peak occurs because these two influential users further ... followers.”

Reviewer: 2

Comments to the Author(s)

In “Why cannot long-term cascade be predicted? Exploring temporal dynamics in information diffusion processes”, Xiao-ke Xu, Ren-meng Cao and Xiao-fan Liu explore long-term and short-term prediction of the popularity of tweets on the social media platform Weibo. They use a machine learning model which bases its predictions on the characteristics of the Weibo tweet, features of the tweet’s early retweeting record, and the characteristics of the user who posted the tweet. I have little experience in machine learning or tweet dynamics, and cannot comment on whether the machine learning model and training process are identified unambiguously. To me, the paper seemed well-motivated and well-organised, with conclusions largely justified by the data and supporting analysis. I recommend some small changes before publication.

Conclusions and analysis:

In their current form, some conclusions are not entirely supported by the data and analysis. In such cases, I suggest either adjusting the conclusion or extending the analysis.

(1) It is stated that early popularity dynamics are the fastest to lose their predictive power (p. 7, lines 57-59) and the first to lose their predictive power (p. 8, lines 9-11). In my opinion, statements about the rapidity with which predictive power is lost might be misleading, and should probably be modified. I suggest this because the predictive power is only shown at two extreme prediction horizons, and it is hard to infer how prediction performance varies between these two extremes.

(2) In Section 4 (d) it is shown that a model trained on 10000 tweets from 2015 can predict 3000 tweets from 2015 more accurately than it can predict 3000 tweets from 2016, and this result is used to argue the presence of feature drift. I suggest indicating whether the 2015 training and test data are kept separate. If the 2015 training and test data share tweets then the higher accuracy for 2015 could be partly due to a tendency of in-sample prediction error to be lower. In this case, more discussion or analysis may be needed to support the conclusion about feature drift.

(3) In the final paragraph of the conclusion it is suggested that the information diffusion process is generally controllable. It is suggested that information diffusion can be enhanced by having a key opinion leader retweet the information, which is reasonable. However, the statement about controllability seems to suggest the existence of mechanisms to either enhance or suppress information diffusion. Therefore, I suggest also indicating a way in which information diffusion could be suppressed or adjusting the statement about controllability.

Stylistic and typographical points:

The paper contains a small number of typographical errors, and at points it might be possible to enhance its style or clarity.

(5) I found the following sentences (spanning two paragraphs) in the introduction jarring: "In this case, the features constructed based on the early retweeting dynamics will be invalid in the prediction task.

Why does such failure happen?"

I found these sentences confusing because they appear to involve answering the main question of the paper (Why does long term tweet popularity prediction fail?) immediately *before* the question is asked. I suggest modifying or reordering at least one of these two sentences or paragraphs.

(6) I suggest defining "cascade size" and "retweet tree" in the context of tweets before or as the terms are first used (currently, p. 4, lines 15-17). I suggest including a reference for PageRank where the centrality measure is first mentioned (currently, p. 5, line 51). I suggest defining "publisher" before or as the term is first used (currently, p. 5, line 51).

(7) Page 5, line 51: For consistency with the preceding four lines, I suggest changing "publisher" to "posting user" (assuming that the meaning of the terms is the same).

(8) Page 6, line 12: For clarity, I suggest changing "where t represents the length of the observation period." to "where t represents the length of the observation period in minutes."

(9) Page 6, line 57: Please clarify the connection between cascade size cascade_t at time t and number $N_m(t)$ of retweets of tweet m at time t.

(10) Figures 4-5: I suggest changing "Observarion" to "Observation". I also suggest checking that the caption and legend are consistent – the legend and caption respectively indicate that the blue and red curve show the real values.

(11) Figure 6: I suggest concluding the caption with a full stop.

(12) Page 12, lines 19-20: Please check this definition of feature drift: "Feature drift occurs when the validity of features in the data stream changes over time, leading the ML algorithms to ignore irrelevant features and consider the newly relevant ones [39]." This seems to suggest that feature drift refers to a self-updating skill possessed by machine learning models. I would have found the following definition of feature drift more intuitive: "Feature drift occurs when the validity of features in the data stream changes over time, requiring ML models to be retrained to ignore irrelevant features and consider the newly relevant ones [39]."

Final comment:

(13) The authors hypothesise that burstiness and the variability of the original time of posting are two sources of long term unpredictability. This is reasonable, and it is shown that cascade time series are bursty and that the tweet rate varies with time of day. The hypotheses are not proved, but are also not presented as having been proved, so this does not invalidate the conclusions of the paper. However, in my opinion, the paper would be enhanced by extending the analysis slightly to test these hypotheses about the sources of long term unpredictability. The hypotheses could be validated, for example, by examining how prediction performance varies with burstiness or the original time of posting. Perhaps more robustly, the authors could instead train and test one model on time series with burstiness above a certain threshold, train and test another on time series with burstiness below the threshold, and compare the performance of the two models. Analogously, the authors could train and test models using only data from tweets first

posted at different intervals of the day (for example, midnight – 6am, 6am – 12nn, 12nn – 6pm, 6pm-midnight), and compare the performance of the models restricted to different intervals.

Reviewer: 3

Comments to the Author(s)

Report on "Why can't long-term cascade be predicted? Exploring temporal dynamics in information diffusion processes" by R-M Cao, X-F Liu, X Xu

The manuscript demonstrates problems encountered with long-term prediction of Twitter like information networks. The authors demonstrate that short-term prediction of information diffusion in such networks is a manageable task but outside a pre-defined prediction window the machine learning algorithm fails to give reasonable estimates for the retweet popularity.

The manuscript is well written, contains the necessary information and contains informative as well as good designed figures. Different retweet scenarios are well illustrated and while I am not the biggest fan of the bullet like listing (see two lists on page 3 and 5), I believe this is within the freedom of the authors.

The methodology developed is based on the popular XGBoost algorithm and RMSE is used as a goodness measures of the ML algorithm. As a training set 80% of the data set (Weibo data from 2015) is used and the remaining 20% are the test data. In principle the result shown is not surprising. XGBoost is a tree based model and such methods partition the input space of any given problem in a way that makes them largely unable to extrapolate target values beyond the limits of the training data when used for predictions. Given that the burst dynamics reported on around figure 2 is a rare event, it is not surprising that a tree-based model fails to predict it. Given that this is the main point of the manuscript and it is a well known fact in data science, I do not rate the novelty of this manuscript highly.

The manuscript in addition focusses on a body of research dealing with the analysis of twitter-like data. I do believe that this is a too narrow view and research on prediction in social systems like for example Sornette, D.; Deschatres, F.; Gilbert, T.; Ageon, Y (2004). "Endogenous Versus Exogenous Shocks in Complex Networks: an Empirical Test Using Book Sale Ranking". *Physical Review Letters*. 93 (22): 228701. doi:10.1103/physrevlett.93.228701 would give a different interesting perspective to the research area.

===PREPARING YOUR MANUSCRIPT===

===PREPARING YOUR REVISION IN SCHOLARONE===

<https://royalsociety.org/journals/authors/author-guidelines/#data>. You should ensure that

you cite the dataset in your reference list. If you have deposited data etc in the Dryad repository, please include both the 'For publication' link and 'For review' link at this stage.

Author's Response to Decision Letter for (RSOS-202245.R0)

See Appendix A.

RSOS-202245.R1 (Revision)

Review form: Reviewer 1

Is the manuscript scientifically sound in its present form?

Yes

Are the interpretations and conclusions justified by the results?

Yes

Is the language acceptable?

Yes

Do you have any ethical concerns with this paper?

No

Have you any concerns about statistical analyses in this paper?

No

Recommendation?

Accept as is

Comments to the Author(s)

This paper has been revised correctly, and the acceptance is considerable.

Review form: Reviewer 2

Is the manuscript scientifically sound in its present form?

Yes

Are the interpretations and conclusions justified by the results?

Yes

Is the language acceptable?

Yes

Do you have any ethical concerns with this paper?

No

Have you any concerns about statistical analyses in this paper?

No

Recommendation?

Accept as is

Comments to the Author(s)

The authors have done a thorough job of addressing my concerns. Furthermore, compared to the original submission, the revised manuscript offers new, robustly supported insights about the origins of the unpredictability of tweets. I am happy to support publication.

Decision letter (RSOS-202245.R1)

Dear Dr Xu,

It is a pleasure to accept your manuscript entitled "Why cannot long-term cascade be predicted? Exploring temporal dynamics in information diffusion processes" in its current form for publication in Royal Society Open Science. The comments of the reviewer(s) who reviewed your manuscript are included at the foot of this letter.

You can expect to receive a proof of your article in the near future. Please contact the editorial office (openscience@royalsociety.org) and the production office (openscience_proofs@royalsociety.org) to let us know if you are likely to be away from e-mail contact – if you are going to be away, please nominate a co-author (if available) to manage the proofing process, and ensure they are copied into your email to the journal.

Please see the Royal Society Publishing guidance on how you may share your accepted author manuscript at <https://royalsociety.org/journals/ethics-policies/media-embargo/>. After publication, some additional ways to effectively promote your article can also be found here

<https://royalsociety.org/blog/2020/07/promoting-your-latest-paper-and-tracking-your-results/>.

on behalf of Professor Marta Kwiatkowska (Subject Editor)
openscience@royalsociety.org

Reviewer comments to Author:
Reviewer: 1
Comments to the Author(s)

This paper has been revised correctly, and the acceptance is considerable.

Reviewer: 2
Comments to the Author(s)
The authors have done a thorough job of addressing my concerns. Furthermore, compared to the original submission, the revised manuscript offers new, robustly supported insights about the origins of the unpredictability of tweets. I am happy to support publication.

Appendix A

Dear editor,

We would like to thank you, the associate editor, and reviewers for furnishing us with very detailed and helpful reviews. We have made a thorough revision to the manuscript according to the reviewers' comments and suggestions carefully. Moreover, we have expended further effort to remove (or correct) any errors and unclear descriptions presented in the previous manuscript (Paper ID: RSOS-202245). We believe that the results are compelling, and the fundamental message of this manuscript will be of great interest to the readership of *Royal Society Open Science*. In the following, we respond directly to the comments provided by the reviewers and describe the changes we have made to the manuscript.

Responds to the reviewer's comments:

Reviewer 1

General comments: In this manuscript, the author aimed to explore the possible causes of difficulties in predicting information diffusion in the long run. Comparing the predictive ability (and loss of it) of various feature sets, the authors found that early diffusion dynamics are the most predictive ones in short-term prediction tasks but lose most of their predictive power in long-term tasks. Afterward, the authors further dug out two causes that can break the correlation between early and future popularity. Overall, the paper is well-structured and innovative. I recommend publication after some minor revision.

Response: We would like to sincerely thank the reviewer for the careful inspection and valuable suggestions. We have made our best effort to revise the manuscript accordingly.

Question 1-1: "In Section 1. Introduction, the author needs to provide more literature about the impacts of the busy diffusion of online content on popularity prediction and a detailed description about how prior studies solve the difficulty in long-term prediction, so as to put forward the research question."

Response to 1-1: Thanks for the reviewer's comment, which makes our manuscript clear. We have added detailed literature reviews about the impact of the busy diffusion of online content on popularity prediction and a detailed description of how prior studies solve the difficulty in long-term prediction. References 20 to 27 are newly supplemented ones (Page 3, Section 1. Introduction).

Question 1-2: "I suggest that " This dataset contains an original tweeter ID and a list of user IDs the original tweeter follows" in Section 2 Data description and preprocessing might be revised into "original tweeter followed"."

Response to 1-2: Thank you for the suggestion. We have revised the phrase accordingly (Page 4, Section 2. Data description and preprocessing). The revised text is shown as follows:

"The lists of users that each of the posting users and retweeters followed, including 7,977,942 users and 700,434,403 following relations."

Question 1-3: “In Section 3 Following network structure, “mutual friends” may be revised into “mutual following.”

Response to 1-3: We have revised this phrase and other appearances in the manuscript accordingly (Page 5, Section 3(a) Following network structure). The revised descriptions of features are shown below:

“all_degree_F: Number of people who follow or are followed by the posting user.
bi_degree_F: Number of people who follow and are followed by the posting user.
pagerank_F: PageRank centrality in G_F.”

Question 1-4: “In Section 3 Retweeting network structure, I am confused about the description of all_degree_R. In the retweeting network, the relationships between users should be retweet or tweet, not a friend.”

Response to 1-4: We are sorry for the unclear description of all_degree_R. We have clarified the descriptions of all_degree_R and some other features on Page 5, Section 3(a) Retweeting network structure. The changes are also reflected in Table 1. The specific changes are shown below:

“bi_degree_R: Number of people who retweet and are retweeted by the posting user.
all_degree_R: Number of people who retweet or are retweeted by the posting user.
out_degree_R: Number of people who retweet the posting user's tweets.
in_degree_R: Number of people retweeted by the posting user.
pagerank_R: PageRank centrality in G_R.”

Question 1-5: “In Section 3 Early retweet dynamics, the author constructed four types of features instead of three.”

Response to 1-5: We are very sorry for the typo. We have corrected this typo on Page 5, Section 3(a) Early retweet dynamics. The amended text is as follows:

“To capture the trends and fluctuations of cascade sizes over time, four types of temporal features are constructed.”

Question 1-6: “In Section 3 Early retweet dynamics, “cascade 60” might be revised into “cascade t”.”

Response to 1-6: We have corrected the symbol accordingly on Page 5, Section 3(a) Early retweet dynamics, as follows:

“burstiness: this index measures the burst of the popularity time series in the observation period, which is defined as

$$\text{burstiness} = \text{peak} / \text{cascade}_t,$$

where *peak* represents the largest net increase of the retweet time series in the observation period.”

Question 1-7: “In Section 3 tweet content, I am confused about the text “The texts bare the richest information about individual tweets.”

Response to 1-7: We are very sorry for the unclear expression. To avoid further confusion, we decide to remove this sentence (Page 6, Section 3(a) Tweet content) in the revised version.

Question 1-8: “In Figure 1, there are some typos in the caption, “the blue and short lines” might be revised into “the blue and red lines””

Response to 1-8: We have corrected the typo accordingly (Page 7, Section 3(b), Figure 1). The revised caption is shown below:

“Prediction performance and the loss of predictive power of different feature groups in short-term and long-term cascade prediction. The blue and red lines show the prediction performance of each group of features, and all features in the short-term and long-term ... Content denotes the features of original tweets' content features. ALL denotes the collection of all the 79 features.”

Question 1-9: “In Section 4(a), I am confused about the expression “Bursts can appear with very rare probability and hence break the correlation between early-stage popularity and long-term popularity in information diffusion.” In this section, the authors should focus on the effect of burst on the spread of information, not the probability of burst.”

Response to 1-9: We are very sorry for the unclear description of bursts. In the revised manuscript, we have removed this confusing expression (Page 8, Section 4(a), para. 1). According to the reviewer’s suggestion, we focus on examining the impact of the burst dynamics of popularity evolution in the spread of information. Several prior works (References 11, 12, and 19) showed that the burst nature of popularity evolution could break the correlation between early and future popularity dynamics. Based on this, we further analyzed the effect of three different types of burst dynamics of popularity evolution on long-term prediction. The specific experimental results and the corresponding discussion have added in Section 4(a), as follows:

“To test the impact of burstiness to the cascade size predictability, we compared the prediction errors between original tweets that experienced strong, weak, and no burst in the cascade ... In the end, 38% of the original tweets are tagged with strong bursts, 48% with weak bursts, and 14% with no burst.

The prediction performance, i.e., RMSE, for original tweets that experienced strong, weak, and no burst at different times in their cascades ... the original tweets that experienced strong bursts have the lowest prediction performance, while the less severe or fewer bursts an original tweet experiences, the easier it is to predict its cascade size. Therefore, the strength of the bursts indeed leads to the (un)predictability of information cascades” (Page 8, Section 4(b), para. 4-5)

Question 1-10: “In Section 4(b), the authors might give a definition of high/low user activity periods.”

Response to 1-10: Thank you for the valuable suggestion, which made our manuscript clearer. We characterize the hourly variations of user activity by the number of retweets at different moments in a day. Based on the result shown by Fig. 4(a) in the revised manuscript, user activity changes over time; that is, users are more inclined to post original tweets and retweet in the daytime rather than at midnight. Therefore, we consider 0:00 a.m. to 8:00 a.m. as the low activity period and 8:00 a.m. to 24:00 p.m. as the high activity period. The revised text is as follows:

“As shown in Fig. 4(a), user activity changes over time; that is, users are more inclined to post original tweets and retweet in the daytime rather than at midnight. In this study, we consider 0:00 a.m. to 8:00 a.m. as the low activity period and 8:00 a.m. to 24:00 p.m. as the high activity period.”

(Page 11, Section 4(c), para. 1)

Question 1-11: “In Section 4(c), the authors might label the key nodes in the network in Figure 6, so as to give a clear description of the information diffusion process. Besides, the description of Figure 6 is not consistent with the information the figure conveys. For example, the text “The third peak occurs because these two influential users further ... followers.””

Response to 1-11: Thank you for the reviewer’s comment. We have labeled the key nodes with capitalized letters. In the revised manuscript, Figure 6 in the previous version has been revised to Figure 3. We have also revised the descriptions in its caption and the main text (Page 10, Section 4(b) Hard-to-predict dynamics in multi-peak propagations). The revised text is as follows:

“Figure 3. (a) Cumulative growth curves of the real and predicted values of an original tweet with multi-peak propagation. The red curve is the predicted values starting from the 75th minute with 120-minute increments. For each prediction target, we retrain the model. (b) The retweeting network of a multiple-burst tweet. (c) The cascade size time series of a multiple-burst tweet information cascades. The nodes with labels are the most influential users in the retweeting network ...” (Page 10, Section 4(b), Figure 3)

“In the first peak, node A contributes to the major number of retweets. The second peak is caused by two influential users (i.e., nodes B and C). The two influential users spread the information cascades to their followers, respectively, forming two new clusters. The third peak occurs because the two influential users (i.e., nodes B and C) promote the spread of information cascades once again. Besides, two new influential users (i.e., nodes D and E) appear and further improve the diffusion of information cascades.” (Page 9, Section 4(b), para. 2)

Reviewer 2

General comments: In “Why cannot long-term cascade be predicted? Exploring temporal dynamics in information diffusion processes”, Xiao-ke Xu, Ren-meng Cao and Xiao-fan Liu explore long-term and short-term prediction of the popularity of tweets on the social media platform Weibo. They use a machine learning model which bases its predictions on the characteristics of the Weibo tweet, features of the tweet’s early retweeting record, and the characteristics of the user who posted the tweet. I have little experience in machine learning or tweet dynamics and cannot comment on whether the machine learning model and training process are identified unambiguously. To me, the paper seemed well-motivated and well-organised, with conclusions largely justified by the data and supporting analysis. I recommend some small changes before publication.

Response: We would like to sincerely thank the reviewer for the careful inspection and valuable suggestions. We have made our best efforts to revise the manuscript according to your suggestions.

Conclusions and analysis: In their current form, some conclusions are not entirely supported by the data and analysis. In such cases, I suggest either adjusting the conclusion or extending the analysis.

Response: Thanks for the reviewer's comment. There are indeed some conclusions that are not entirely supported by the data and analysis. This problem is mainly caused by the lack of experimental results and our unclear expression. For example, In Section 4, we hypothesized that burstiness and variability of the original time of posting are two sources of long-term unpredictability and directly summarized two scenarios in which the actual tweet popularity largely diverges from the predicted scales. Obviously, it is far-fetched and unconvincing. These hypotheses need to be proved or presented as having been provided. Besides, some misleading descriptions of prediction performance of early popularity dynamics (Page 7, lines 57-59; Page 8, lines 9-11) and the construction of training sets for testing the phenomenon of feature drift (Section 4 (d) Feature temporal drifting) further made reviewer confused and unconvinced about the conclusions.

In the revised manuscript, we have verified the above hypotheses (Page 8, Section 4(a), para. 5; Page 10, Section 4(b), para. 2). For the burstiness of popularity evolution, we set up three control groups to prove the impact of burstiness on long-term cascade prediction: *strong burst*, *weak burst*, and *no burst*, finding that the original tweets that experienced strong bursts have the lowest prediction performance, while the less severe or fewer bursts an original tweet experiences, the easier it is to predict its cascade size. That is to say, the strength of the bursts indeed leads to the (un)predictability of information cascades. As for the variability of the original time of posting, we clearly define low/high user active period and divide the original tweets in the testing set into two parts by their posting time: *high activity* and *low activity*. The prediction results from the two types of samples show that original tweets posted during the low activity period may experience unexpected cascading behavior and diverge from prediction severely.

In the following comments (from **Question 2.1 to 2.12**), we will provide a point-by-point response to each of the comments in detail. Thanks for the reviewer's critical comments, which made our manuscript clearer.

Stylistic and typographical points: The paper contains a small number of typographical errors, and at points, it might be possible to enhance its style or clarity.

Response: We are very sorry for the stylistic and typographical errors. In addition to the suggestions

proposed by the reviewer (**Question 2-4 to 2-12**), we have expended further effort to remove or correct any errors or unclear descriptions presented in the previous manuscript.

Question 2-1: *“It is stated that early popularity dynamics are the fastest to lose their predictive power (Page 7, lines 57-59) and the first to lose their predictive power (Page 8, lines 9-11). In my opinion, statements about the rapidity with which predictive power is lost might be misleading, and should probably be modified. I suggest this because the predictive power is only shown at two extreme prediction horizons, and it is hard to infer how prediction performance varies between these two extremes.”*

Response to 2-1: We are very sorry for the misleading statement about the rapidity with which predictive power is lost. It is indeed hard to infer how prediction performance varies between these two extremes only by the two extreme prediction horizons. In the revised manuscript, we have corrected this problem. The amended text is as follows:

“In the long-term prediction, the RMSEs of all feature groups substantially increase, and the most informative group of features. In a word, features of early popularity dynamics not only are the key predictors of cascade size but also the fast to lose their prediction ability lose their predictive power most severely from short-term to long-term prediction.” (Page 7, Section 3(b), para. 2)

Question 2-2: *“In Section 4 (d) it is shown that a model trained on 10000 tweets from 2015 can predict 3000 tweets from 2015 more accurately than it can predict 3000 tweets from 2016, and this result is used to argue the presence of feature drift. I suggest indicating whether the 2015 training and test data are kept separate. If the 2015 training and test data share tweets then the higher accuracy for 2015 could be partly due to a tendency of in-sample prediction error to be lower. In this case, more discussion or analysis may be needed to support the conclusion about feature drift.”*

Response to 2-2: We are very sorry that there is a lack of the statement about the independence between training and test data in Section 4(d). In this section, we train a model based on 1000 tweets from 2015 and predict 3000 tweets from 2015 and 3000 tweets from 2016 to check the presence of feature drift. The 2015 training and test data are kept separate, and there is no overlap between the two datasets. That is to say, the experimental result is reliable and not caused by a tendency of in-sample prediction error to be lower. In the revised manuscript, we have indicated that the two datasets are independent of each other. The added text is shown as follows:

“To confirm the existence of feature drift in cascade propagation prediction, a predictive model is constructed and trained based on 10,000 original tweets published in 2015, and 3000 original tweets published in 2015 and 2016 are randomly selected for testing, respectively. There is no overlap between the train data and the test data.”

Question 2-3: *“In the final paragraph of the conclusion it is suggested that the information diffusion process is generally controllable. It is suggested that information diffusion can be enhanced by having a key opinion leader retweet the information, which is reasonable. However, the statement about controllability seems to suggest the existence of mechanisms to either enhance or suppress information diffusion. Therefore, I suggest also indicating a way in which information diffusion could be suppressed or adjusting the statement about controllability.”*

Response to 2-3: We are very sorry for the unclear statement about the controllability of the spread

of information in Section 5. We have revised the statement and mentioned the suppression of diffusion about the controllability, as follows:

“The controllability embodies in two ways: micromanagement during the diffusion process, e.g., the introduction, or the suppression, of a key opinion leader, which can boost an already started diffusion process or preventing undesired future diffusion, and changing the underlying online environment can also alter the information diffusion mechanism, hence information popularity in the new environment.” (Page 13, Section 5, para. 2)

Question 2-4: “I found the following sentences (spanning two paragraphs) in the introduction jarring:

“In this case, the features constructed based on the early retweeting dynamics will be invalid in the prediction task.

Why does such failure happen?”

*I found these sentences confusing because they appear to involve answering the main question of the paper (Why does long term tweet popularity prediction fail?) immediately *before* the question is asked. I suggest modifying or reordering at least one of these two sentences or paragraphs.”*

Response to 2-4: We are very sorry for the jarring and confusing sentences in Introduction. We have reorganized the logical order of the section of Introduction. Besides, we have also provided more literature about the impact of the busy diffusion of online content on popularity prediction and a detailed description of how prior studies solve the difficulty in long-term prediction to clearly presenting our research question (Page 3, Section 1. Introduction, para. 2-3).

Question 2-5: “I suggest defining “cascade size” and “retweet tree” in the context of tweets before or as the terms are first used (currently, p. 4, lines 15-17). I suggest including a reference for PageRank where the centrality measure is first mentioned (currently, p. 5, line 51).”

Response to 2-5: Thanks for the review's comment, which made our manuscript clearer. In this study, we define the cascade size of a piece of information as the number of people who retweeted an original post (Page 3, Section 1. Introduction, para. 4). “Retweet tree” has been replaced with “retweeting network”. The reference for the PageRank algorithm has been added in the manuscript, which is numbered as 36. The revised text as follows:

“Specifically, we consider a Twitter-like online social networking service as a typical example of information diffusion and adopt a best-practice ML method to predict the cascade size, i.e., the number of people who retweeted an original post.”

37. Gleich DF. 2015 PageRank Beyond the Web. SIAM Rev. 57, 321–363. (doi:10.1137/140976649)

Question 2-6: “I suggest defining “publisher” before or as the term is first used (currently, p. 5, line 51). Page 5, line 51: For consistency with the preceding four lines, I suggest changing “publisher” to “posting user” (assuming that the meaning of the terms is the same).”

Response to 2-6: Thank you for the comment. “Publisher” has been replaced by “posting user” to refer the user who posted an original tweet. We have also unified some other confusing phrases in the manuscript, as follows:

“*pagerank_R*: the PageRank centrality [37] of the posting user.” (Page 5, Section 3(a), Table 1)

Question 2-7: “Page 6, line 12: For clarity, I suggest changing “where t represents the length of the observation period.” to “where t represents the length of the observation period in minutes.””

Response to 2-7: Thank you for the review’s suggestion. We have revised the sentence accordingly (Page 6, the description of *cascade*). The revised text is as follows:

“*cascade*: the cumulative time series of cascade sizes organized in 1-minute intervals in the observation period ... where t represents the length of the observation period in minutes.”

Question 2-8: “Page 6, line 57: Please clarify the connection between cascade size *cascade_t* at time t and number $N_m(t)$ of retweets of tweet m at time t .”

Response to 2-8: We are very sorry that there is a lack of the statement of the connection between *cascade size*, *cascade_t* at time t , and number $N_m(t)$ of retweets of tweet m at time t . The three concepts are essentially the same. Cascade size refers to the number of people who retweeted an original post. *Cascade_t* denotes the number of people who retweeted the original post when it was spread at time t . $N_m(t)$ represents the actual cascade size of the original post at time t . In the revised manuscript, we have given a clear description of the three concepts, as follows:

“Specifically, we consider a Twitter-like online social networking service ... to predict the cascade size, i.e., the number of people who retweeted an original post.” (Page 2 Section 1, para. 4)

“*cascade*: the cumulative time series of cascade sizes organized in 1-minute intervals in the observation period... where t represents the length of the observation period in minutes. *cascade_t* denotes the number of people who retweeted the original post when it was spread at time t .” (Page 6 Section 3(a) retweeting network feature)

“We use the Root Mean Square Error (RMSE) to evaluate the predictive performance of our model ... the better predictive performance we achieve. Let $N_m(t)$ be the real cascade size of the original tweet m at time t .” (Page 6, Section 3(b), para. 2)

Question 2-9: “Figures 4-5: I suggest changing “Observarion” to “Observation”. I also suggest checking that the caption and legend are consistent – the legend and caption respectively indicate that the blue and red curve show the real values.”

Response to 2-9: We are sorry for the typos in Fig. 4 and Fig. 5. In the revised manuscript, Fig. 4 and Fig. 5 in the previous version have been revised to Fig. 3(a) and Fig. 4(b), respectively. We have amended these typos and rearranged the figures appropriately. The revised text is as follows:

“Figure 3. (a) Cumulative growth curves of the real and predicted values of an original tweet with multi-peak propagation. The red curve is the predicted values starting from the 75-th minute with 120-minute increments. For each prediction target, we retrain the model.” (Page 10, Figure 3(a))

“Figure 4. (a) The number of retweets at different moments in day in our dataset. Each bar represents the number of retweets in an hour. (b) The real and predicted cascade sizes of an original tweet published around five o’clock. The red curve is the predicted values starting from the 75th minute with 120-minute increments. For each prediction target, we retrain the model.” (Page 11, Figure 4(b))

Question 2-10: *“Figure 6: I suggest concluding the caption with a full stop.”*

Response to 2-10: We have revised the manuscript accordingly. Fig. 6 in the previous version has been revised to Fig. 3(b) and Fig. 3(c) in the revised manuscript. The revised text is shown below:

“Figure 3 (a) Cumulative growth curves of the real and predicted ... (c) The cascade size time series of a multiple-burst tweet information cascades. The nodes with labels are the most influential users in the retweeting network.” (Page 10 Figure 3(c))

Question 2-11: *“Page 12, lines 19-20: Please check this definition of feature drift: “Feature drift occurs when the validity of features in the data stream changes over time, leading the ML algorithms to ignore irrelevant features and consider the newly relevant ones [39].” This seems to suggest that feature drift refers to a self-updating skill possessed by machine learning models. I would have found the following definition of feature drift more intuitive: “Feature drift occurs when the validity of features in the data stream changes over time, requiring ML models to be retrained to ignore irrelevant features and consider the newly relevant ones [39].”*

Response to 2-11: Thank you very much for the suggestion. We have revised the manuscript accordingly, as follows:

“Feature drift occurs when the validity of features in the data stream changes over time, requiring ML models to be retrained to ignore irrelevant features and consider the newly relevant ones [46].

Barddal et al. analyzed the impact of feature drift on some ML algorithms and found that the existing ML algorithms did not solve the problem of feature drift [48].” (Page 12, Section 4(d), para. 1)

Question 2-12: *“The authors hypothesise that burstiness and the variability of the original time of posting are two sources of long term unpredictability. This is reasonable, and it is shown that cascade time series are bursty and that the tweet rate varies with time of day. The hypotheses are not proved, but are also not presented as having been proved, so this does not invalidate the conclusions of the paper. However, in my opinion, the paper would be enhanced by extending the analysis slightly to test these hypotheses about the sources of long term unpredictability. The hypotheses could be validated, for example, by examining how prediction performance varies with burstiness or the original time of posting. Perhaps more robustly, the authors could instead train and test one model on time series with burstiness above a certain threshold, train and test another on time series with burstiness below the threshold, and compare the performance of the two models. Analogously, the authors could train and test models using only data from tweets first posted at different intervals of the day (for example, midnight – 6am, 6am – 12nn, 12nn – 6pm, 6pm-midnight), and compare the performance of the models restricted to different intervals.”*

Response to 2-12: The reviewer has indeed raised an important issue and suggested an excellent direction for improving our study. We have compared the difference of the prediction errors (RMSE) between samples (retweet cascades) with different levels of burstiness, numbers of peaks, and the variability of original posting times. The models are retrained with each group of stratified samples. Please see the revised Section 4 for details. The newly added text is as follows:

“To test the impact of burstiness to the cascade size predictability, we compared the prediction errors between original tweets that experienced strong, weak, and no burst in the cascade. An original

tweet is considered to experience a strong burst if its largest peak height is higher than the average height of all peaks of all the original tweets' cascades. Other original tweets with at least one burst are considered experienced weak bursts. In the end, 38% of the original tweets are tagged with strong bursts, 48% with weak bursts, and 14% with no burst.

The prediction performance, i.e., RMSE, for original tweets that experienced strong, weak, and no burst at different times in their cascades are shown in Fig. 2(b) ... Therefore, the strength of the bursts indeed leads to the (un)predictability of information cascades.” (Page 8, Section 4(a), para. 4-5)

“Similar to the previous section, we stratified the original tweets that experienced at least one burst into those which experienced only one burst and those which experienced multiple bursts ... Fig. 3(d) shows that it is easier to predict the cascade size for original tweets that experienced only one peak in their cascades. This finding confirms that the complex burstiness in information cascades hinders the predictability of the cascade process.” (Page 10, Section 4(b), para. 4)

“To further test the difference in predictability for cascades that started in high and low activity periods, respectively, we stratified the original tweets by their posting time into two groups ... This finding confirms that original tweets that are posted during low activity period may experience unexpected cascading behavior and diverge from prediction severely.” (Page 11, Section 4(c), para. 3)

Reviewer 3

General comments: "The manuscript demonstrates problems encountered with long-term prediction of Twitter like information networks. The authors demonstrate that short-term prediction of information diffusion in such networks is a manageable task but outside a pre-defined prediction window the machine learning algorithm fails to give reasonable estimates for the retweet popularity. The manuscript is well written, contains the necessary information and contains informative as well as good designed figures.

Response: We would like to thank the reviewer for the careful inspection and valuable suggestions to our manuscript. We have made our best effort to revise the manuscript accordingly.

Question 3-1: Different retweet scenarios are well illustrated and while I am not the biggest fan of the bullet like listing (see two lists on page 3 and 5), I believe this is within the freedom of the authors.

Response to 3-1: Thank you for the reviewer's suggestion. The two lists on page 3 and 5 are used to give a brief description of the following and retweeting network features. Given that these features do not involve complex formulas and have been summarized in Table 1, we agree with the reviewer's suggestion and have removed some of the bullet like listings (Page 3, Section3(a) Following network structure; Page 5, Section 3(a) Retweeting network structure).

Question 3-2: "The methodology developed is based on the popular XGBoost algorithm and RMSE is used as a goodness measures of the ML algorithm. As a training set 80% of the data set (Weibo data from 2015) is used and the remaining 20% are the test data. In principle the result shown is not surprising. XGBoost is a tree-based model and such methods partition the input space of any given problem in a way that makes them largely unable to extrapolate target values beyond the limits of the training data when used for predictions. Given that the burst dynamics reported on around figure 2 is a rare event, it is not surprising that a tree-based model fails to predict it. Given that this is the main point of the manuscript and it is a well-known fact in data science, I do not rate the novelty of this manuscript highly."

Response to 3-2: Sorry for the confusion brought by "rare event." We have rephrased it as burstiness is instead a common phenomenon in retweet cascades. Specifically, 86% of the cascades have experienced at least one burst during their lifetime. The main point of this paper is to explore the reasons why information diffusion cannot be accurately predicted when the time span between training data and prediction target is long. We first showed that using the early diffusion pattern of a piece of information, one can use ML models to predict the diffusion at any future time. However, the predictability of the near future is relatively higher than that of the distant future. Then, we examined and compared (according to the suggestion from Reviewer 2) the predictability of samples (retweet cascades) with differential values in some of the features, i.e., burstiness and posting time to be specific. We have shown that these features do cause the predictability of information diffusion to decrease, especially for long-term prediction (See **Response 2-12** for the newly added results and text content). The contribution of our paper is some of the causes that break the correlation between early and future popularity in information diffusion.

Question 3-3: "The manuscript in addition focusses on a body of research dealing with the analysis of twitter-like data. I do believe that this is a too narrow view and research on prediction in social

systems like for example Sornette, D.; Deschatres, F.; Gilbert, T.; Ageon, Y (2004). "Endogenous Versus Exogenous Shocks in Complex Networks: an Empirical Test Using Book Sale Ranking". *Physical Review Letters*. 93 (22): 228701. doi:10.1103/physrevlett.93.228701 would give a different interesting perspective to the research area.”

Response to 3-3: We agree with the reviewer's comment. The research framework in this study can not only be applied to the information dissemination of social networks such as Twitter, Facebook, and Weibo, but also other fields such as book sales recommendation and citation impact of scientific papers. Thank you very much for recommending this important literature to us in this field. We have carefully read the enlightening literature and cited it in the revised manuscript. The reference number of the enlightening literature is 32. The added text is as follows:

“We compare the predictive power of each group of features in both short-term and long-term prediction tasks as well as the loss of indicativeness of these features ... Our findings further enhance the comprehension of the information diffusion process and may assist in the control of such process. Besides, our research method can not only be applied to the information dissemination of social networks such as Twitter, Facebook, and Weibo, but also other fields such as book sales recommendation [32] and citation impact of scientific papers [33]” (Page 3, Section 1, para. 5)

32. Sornette D, Deschâtres F, Gilbert T, Ageon Y. 2004 Endogenous Versus Exogenous Shocks in Complex Networks: An Empirical Test Using Book Sale Rankings. *Phys. Rev. Lett.* 93. (doi:10.1103/physrevlett.93.228701)

Finally, in addition to our responses to the above comments, and our corresponding revisions, we have expended further effort to correct any writing errors or unclear descriptions presented in the previous manuscript. We hope that the above responses to each of the concerns of the editor and reviewers will be sufficient to convince you of the importance of our results. We believe that the results are compelling and that the fundamental message of this manuscript will be of great interest to the readership of *Royal Society Open Science*.

Ren-Meng Cao and Xiao-Ke Xu (xuxiaoke@foxmail.com)
Dalian Minzu University

Xiao Fan Liu
City University of Hong Kong